# Effects of Potassium Deficiency on the Growth of Tea (*Camelia sinensis*) and Strategies for Optimizing Potassium Levels in Soil: A Critical Review

Wei Huang [1], Minyao Lin [1], Jinmei Liao [1], Ansheng Li [1], Wugyan Tsewang [2], Xuan Chen [3], Binmei Sun [1], Shaoqun Liu [1,*] and Peng Zheng [1,*]

1   College of Horticulture, South China Agricultural University, Guangzhou 510642, China;
    huangwei_chris@stu.scau.edu.cn (W.H.); lmy17@stu.scau.edu.cn (M.L.);
    ljm19127614165@stu.scau.edu.cn (J.L.); 1533854642las@stu.scau.edu.cn (A.L.); binmei@scau.edu.cn (B.S.)
2   Tibet Autonomous Region Agricultural Technology Extension Service Center, Lhasa 850000, China;
    wjcr1964@163.com
3   Chaozhou Tianxia Tea Industry Co., Ltd., Chaozhou 521000, China; cx07680768@163.com
*   Correspondence: scauok@scau.edu.cn (S.L.); zhengp@scau.edu.cn (P.Z.)

**Abstract:** Potassium is among the three essential macronutrients for tea plants, along with nitrogen and phosphorous, and plays important roles in growth and stress response. Potassium is absorbed by plants in larger amounts than any other mineral element except nitrogen and, in some cases, calcium. At present, more than 59% of China's tea gardens are in a state of potassium deficiency, which negatively affects tea quality and yield. This paper reviews the effects of potassium deficiency on tea plant growth and stress response, details factors affecting potassium supply and demand in tea gardens, examines the interactions between potassium and other elements in soils, and provides strategies for optimizing potassium levels in soils. Potassium is positively correlated with the elements nitrogen, copper, and zinc. Sufficient potassium dramatically improves the yield and quality of tea: it accelerates metabolism, promotes synthesis of catechins, and strengthens biotic and abiotic resistance by activating and regulating different enzymes. Moderate application of potassium fertilizers, along with potassium-solubilizing bacteria, can regulate the ratio of different forms of potassium and increase available potassium in soils of tea gardens. We suggest that research on potassium occurring in soils and its interaction with other elements be strengthened, so as to improve the efficient use of potassium fertilizers in tea gardens and maintain the balance of elements in soils.

**Keywords:** *Camelia sinensis*; potassium; soil; growth

## 1. Introduction

Potassium is an essential macronutrient for plants, including tea (*Camelia sinensis*); and in many tea gardens, potassium is the main limiting growth factor [1]. When soils are slightly deficient in potassium, leaves of the tea plant are smaller, maintain a reduced photosynthesis rate and have a short life. When soils are in severe potassium deficiency, young leaves gradually turn from green to light yellow [2] and become smaller and thinner; dormant buds increase and the internodes shorten. The most obvious feature of severe potassium deficiency is brown spots on leaf tips or along the edge of adult leaves. The spots gradually become brown patches [3] and spread to the veins, and the leaves curl and fall off the plant prematurely in massive amounts. When the third or fourth leaf falls from the tree, the top dies and loses regeneration capacity. In this state, the tea plant is more susceptible to anthrax, tea brown blight, and other diseases [4]; the leaves turn yellow and then brown, bronze pale gray edges form, and the leaf completely dries [5].

The tea plant is a long-lived perennial crop that grows in tropical to subtropical climates; it thrives in humidity and aluminum-rich acid soils [6]. However, high temperatures

and abundant rainfall in tropical to subtropical regions lead to strong soil leaching. Since tea farmers prefer fertilizers that are high in nitrogen and phosphate, potassium is often not sufficiently replenished. These factors, combined with extended periods of harvesting young leaves, which are rich in potassium, have resulted in a decrease in potassium content in tea garden soils [7]. In recent years, there has been an increasing number of studies on potassium nutrition of the tea plant, and it has been proved that potassium fertilizers can significantly improve the yield and quality of tea.

## 2. Characteristics of Potassium Supply and Demand in Tea Gardens

### 2.1. Characteristics of the Potassium Content of the Soils in Tea Gardens

Rapid nutrient consumption of high-yielding tea varieties in intensive crop cultivation systems [8] and the lack of solutions to supplement potassium have caused a serious decline in potassium in soils. The supply of potassium in the tea plant mainly comes from soils, so the potassium content of the soils directly affects plant growth. Potassium availability is affected by various factors: soil texture, soil pH, soil depth and liming material. K is absorbed by plants in larger amounts than any other mineral element except nitrogen and, in some cases, calcium (Table 1). K uptake is often equal to or more than nitrogen uptake. In most cases, the potassium content of fertile soils and soil solutions ranges from 100 to 500 µmol/L [9]. The availability of potassium in soils to plants depends on its concentration and availability (i.e., form).

**Table 1.** Properties of potassium in soil and plants.

| Property | Potassium |
| --- | --- |
| Soil composition | Inorganic cation or component of soil minerals |
| Soil reactions | Simple chemical reactions in soils |
| Plant-uptake mechanism | Taken up only as $K^+$ |
| Plant physiological role | An osmotic regulator |

Ruan et al. [7] evaluated the features of potassium in soils of major tea-producing provinces in China by conducting soil tests and analyzing soil buffering capacity (Q/I relationships). Given the ample rainfall in most tea-producing areas as well as the soils' characteristic low cation exchange capacity, strong acidity, and rich exchangeable aluminum, cations including potassium ions are easily leached. In well-balanced tea gardens, 1 kg of soil contains approximately 10~20 g total potassium, which includes 101~820 mg slow-release potassium (average, 276 mg/kg) (Table 2). More than 59% of tea gardens, mostly in Guangdong, Guangxi, Yunnan and other Chinese southern provinces, have less than 80 mg/kg potassium [10]. The concentration of available potassium in one-third of tea gardens, mainly in hilly red soil regions in southern China, is less than 50 mg/kg. Their conclusion: the potassium content decreases as tea gardens are further south in China [7].

The available potassium content is an index of tea yield. When the potassium content of soil is low, the yield of tea is unsatisfying [1]. According to soil nutrient classification standards of the second national soil survey in China [11], when the available potassium content in soil is less than 100 mg/kg, it is considered as potassium deficiency. Jianyun [12] set the exchangeable potassium content of 80 mg/kg soil as a critical value. It is widely acknowledged that 80 mg/kg is the minimum available potassium concentration required to maintain high-producing tea gardens (Table 2) [13].

**Table 2.** The critical value of K.

| Available K of Soils (mg/kg) | K Content in Adult Leaves (mg/g) | K Content in Twigs (mg/g) | Exogenous Application of K (μmol/L) | References |
|---|---|---|---|---|
| <80 | - | - | - | [1,12,13] |
| <100 | - | - | - | [7,11] |
| - | <8 | <6.85 | <100 | [14] |
| <50 | - | - | - | [15] |
| - | - | - | <100 | [7,14,16–18] |

*2.2. Nutritional Characteristics of Potassium to the Tea Plant*

Nutrient deficiency of plants is often noticed by the resulting growth inhibition. In general, the symptoms of potassium deficiency appear when the potassium content of adult leaves is lower than 8 mg/g, or that of seedling leaves is lower than 6.85 mg/g [14]. When available potassium of soils is below 50 mg/kg, the tea plant will severely lack potassium, which can lead to serious disease [15]. Analysis of soil samples from 54 counties distributed in 16 major tea-producing provinces in China revealed that the exchangeable potassium in approximately 74% of soil samples was below 100 mg/kg [7]. In most tea-producing areas of China, the harvesting season of leaves and buds, which is from spring (usually late February and March) to autumn (September and October), can last for seven to eight months or even longer. Fertilization in spring has a greater effect on the content of nitrogen, phosphorus and potassium than in summer and autumn [19].

**3. Effect of Potassium on Tea Plant Growth**

*3.1. Effect of Potassium on Biomass of the Tea Plant*

Studies have shown that when potassium is inadequate (potassium < 100 μmol/L), the biomass of roots, stems and leaves decreases, while the root–shoot ratio improves [7,14,16–18]. When potassium is adequate, there is an increase in overall biomass [20]. After the potassium concentration reaches a certain extent (800 μmol/L), the dry weights of roots, stems, and leaves plateau [18]. Gong Xuejiao et al. [16] obtained similar results: with an increase in potassium concentration, the biomass of whole plants, twigs, stems and roots increased linearly, reached a peak when the potassium concentration was 682~865 μmol/L, and then slowly decreased. Gong [20] also defined the optimal range of potassium content in adult leaves (10.03~10.83 mg/g) and twigs (17.72~19.11 mg/g), plus the optimal concentration (4.69~5.96 mmol/L) for promoting chlorophyll synthesis and speeding up the net photosynthetic rate of adult leaves. The potassium level is also related to plant age: The preservation of potassium aboveground and the annual absorption increases with age [7].

Potassium fertilizers are necessary for tea gardens in potassium starvation to make the plants grow and thrive. Within the safe dosage range, either potassium chloride (KCl) or potassium sulfate ($K_2SO_4$) can increase yield and quality effectively [21], and the latter can increase the proportion of quality tea [13]. Lei Qiong [22] insisted that potassium fertilizers can not only balance different forms of potassium in soils, but also work with nitrogen and phosphorus to produce massive amounts of good tea. Ruan et al. [7] found that when the exchangeable potassium content in the soil of tea gardens is lower than 80 mg/kg, there is a significant increase in yield after application of $K_2SO_4$ or KCl. The single use of base manure in autumn has similar or even better effects, making it possible to reduce labor costs. The study results also indicated that the yield rises markedly when the dose of KCl or $K_2SO_4$ is 124 or 156 kg/ha, respectively, and it is at the highest level when the dose is 248 or 312 kg/ha, respectively (Table 3). A smaller potassium supply harms tea production, changes the biochemical characteristics of the tea plant, and threatens the nutritional status of tea leaves [23,24].

**Table 3.** Optimal Levels of K in Soil for Tea Crop.

| Site | Time | Soil Conditions | Experimental Conditions | Tree Age | Fertilizer | Optimal Levels of K in Soil | References |
|---|---|---|---|---|---|---|---|
| China | 1990~2010 | - | field experiment | - | KCl and $K_2SO_4$ | 124 kg/ha~160 kg/ha | [7] |
| China | 1999~2004 | - | field experiment | - | $K_2O$ | $K_2O$:MgO = 1:0.15~1:0.25 | [12] |
| Low-hilly red soil tea garden; China | 2004 | Red soil | field experiment | >5-year-old | $K_2SO_4$ and KCl | 20~50 kg/ha | [13] |
| Low-hilly red soil tea garden; China | 2004 | Red soil | field experiment | <5-year-old | KCl | <225 kg/ha | [13] |
| Sichuan Academy of Agricultural Sciences; China | 2014 | - | hydroponic experiment | 12-month-old | $K_2SO_4$ | 4.69 mmol/L~5.96 mmol/L | [20] |
| UPASI Tea Experimental Farm; India | 2000~2004 | - | field experiment | - | KCl | N:K = 1:0.83 or 1:0.62 | [25,26] |
| UPASI Tea Experimental Farm; India | 2000~2004 | - | field experiment | - | $K_2SO_4$ | N:K = 1:0.21 or 1:0.42 | [25,26] |

### 3.2. Effects of Potassium on Metabolism in Camelia Sinensis

Potassium facilitates almost all biochemical reactions in plants. The proper amount of potassium strengthens photosynthesis and metabolism of sugars and proteins, which ultimately promotes the production of catechins. On the other hand, suboptimal concentrations of potassium slow the plant's ability to exchange gases by increasing stomatal resistance. This in turn slows the activity of ribulose 1,5-bisphosphate (RuBP) carboxylase, and consequently reduces the net photosynthetic rate. Potassium also maintains the transmembrane proton gradient of chloroplasts and thylakoids in the light and ensures the high pH of chloroplast stroma, which enables photophosphorylation and CO2 assimilation [27].

Potassium deficiency disturbs the synthesis of metabolic enzymes in the tea plant. The activity of catalase (CAT), ascorbate peroxidase (APX) and monodehydroascorbate reductase (MDAR) in tea leaves significantly decreases under potassium deficiency [18]. Potassium deficiency also reduces photosynthetic electron transport capacity and affects the normal progression of photosynthesis. Excessive potassium accelerates the metabolic rate of GA (gallic acid), gallocatechol (GC), catechin (C), epicatechin gallate (ECG), epigallocatechin gallate (EGCG), etc., which is not conducive to catechin accumulation [28].

To a certain extent, potassium and magnesium can stimulate the production of terpenes in tea. Among the acid-hydrolyzed aroma components, the content of oxidized linalool and linalool in tea is higher. Potassium indirectly regulates the synthesis of terpenes by affecting the activity of 3-hydroxy-3-methylglutaryl-coenzyme A (HMG-CoA) reductase. It is only at a certain level of HMG-CoA reductase that more terpenes can be produced. Magnesium also plays a role in synthesizing terpenes. In this process, it assists in metabolism of potassium in the plant as well [29].

After the use of potassium fertilizers, α-cubebene and other isopentenyl-diphosphate volatile substances are increased and geraniol is decreased [30]. The principles behind it are stated as follows: α-cubebene is produced by the mevalonic acid (MVA) pathway. In the second step of this pathway, MVA is produced from HMG-CoA with the help of HMG-CoA reductase. We can speculate that potassium prompts the change in related isopentenyl-diphosphate volatile substances through affecting the activity of this reductase.

Adequate potassium increases the levels of some free amino acids. For example, there is an increase in phenylalanine, which is a precursor to the volatile substances benzaldehyde, benzyl alcohol and phenylethyl alcohol (PEA). The increase in phenylalanine-derived volatile substances in the experiment was possibly due to the use of potassium fertilizers, which stimulate sugar accumulation and the absorption of nitrogen in the tea plant to ultimately increase the content of phenylalanine [30]. Zhou et al. [31] found that a higher level of potassium activates the activity of recombinant theanine synthetase (CsTSI) and raises the ethylamine content (a precursor to L-theanine), and thus improves the production of L-theanine at the roots of the tea plant.

### 3.3. Effect of Potassium on Tea Quality

Tea produced from potassium-deficient soil lacks the intoxicating flavors of tea grown in well-balanced soils [17]. Potassium deficiency in mature tea plants decreases amino acids, caffeine, water extract, and EGCG in tea leaves, while greatly increasing tea polyphenols, catechins, epigallocatechin (EGC), and epicatechin (EC), as well as the phenol/ammonia ratio. Given the lower content of aromatic alcohols, aldehydes, and esters and the weaker diversity of aromatics, such teas lack quality. Some scholars insist that exogenous application of potassium would influence the levels of polyphenols and theaflavins in the tea plant, which would reverse this situation [32].

On the other hand, in young tea plants, potassium has little direct effect on the content of free amino acids and caffeine, and has a positive effect on the content of water extract, catechins, and tea polyphenols [33]. After using potassium fertilizers, the concentration of total free amino acids and extract from tea infusion rises dramatically, while the concentration of polyphenols in autumn tea increases correspondingly [7].

Wei [34] performed experiments to prove that different concentrations of potassium affect the amino acids and aroma contents of different types of tea. Longjing #43 leaves had the highest arginine content when potassium in soil was 20 mg/kg. Arginine declined as the potassium concentration further increased; however, it was still higher than in potassium-deficient conditions. The shuchazao cultivar had outstanding amounts of arginine when potassium was 100 mg/kg in soil. Venkatesan et al. [35] observed that the polyphenol and free amino acid contents increased with increasing potassium concentration. At nitrogen levels of 0 or 50 kg/ha, caffeine content positively correlated with the dose of potassium fertilizers. When nitrogen stabilized at 300 kg/ha/y, the caffeine content increased with the increase in potassium fertilizers, and peaked at a ratio of 1:0.83 [23].

The content of aroma components in tea changes most obviously when potassium is applied as a chloride salt (i.e., potassium chloride). More specifically, fragrant compounds, such as linalool, linalool oxides, benzyl alcohol, β-PEA, nonanal, and 1-octanol, increase significantly, while aroma components with low boiling points and grassy smells (heptanal, leaf alcohol, etc.) decrease. In other words, grassy smells become lighter, while refreshing floral scents become stronger and persist for longer [28]. Wei [34] also found that when the extra amount of exogenous potassium is 40~60 mg/kg in soil, typical terpene fragrances, including trans-linalool oxide, linalool oxide, linalool, geraniol, and nerolidol, reach the maximum value in leaves of Longjing #43 and shuchazao. Venkatesan et al. [23] found that more nitrogen and potassium (NK) fertilizer causes higher concentrations of Group II compounds, which produce pleasant scents. Three of them—linalool, methyl salicylate and benzaldehyde—contribute to the flavor index (FI) in tea.

Given the long interval of time between the use of phosphorus and potassium fertilizers, the concentrations of available phosphorus and potassium in soils during summer and autumn harvests may be lower than in spring time. In order to raise the quality of summer and autumn tea, the timing of fertilizer application, especially that of phosphorus and potassium, should be based on the specific nutrient needs of the tea plant at that time period [19].

### 3.4. Effects of Potassium on Tea's Abiotic and Biotic Stress Response

3.4.1. Drought Resistance

Drought stress is among the main factors limiting tea yield. Potassium exists in plant cells in ionic states, and keeping the potassium in balance is an important strategy for the tea plant to cope with drought stress. When plants are subjected to drought stress, proteins on the cell membrane immediately respond to the stress signal, letting the intracellular potassium ion ($K^+$) rapidly leak out for several minutes or even hours. It is evident that plants' responses and subsequent water supplementation are crucial for drought tolerance, both of which are related to the regulated activity of the plasma membrane proton-pump ATPase ($H^+$-ATPase) [35]. Potassium fertilizers bring more $K^+$ to tea plant cells, and these ions can increase the osmotic pressure of the sap of plant cells, increasing water absorption by roots, stomatal closure, and the stability of biofilms. As a result, the tea plant's drought resistance is greatly enhanced with proper amounts of available potassium [13,22,36–40].

$K^+$ retention is indispensable for reducing drought-induced damage to the tea plant, and exogenous potassium (i.e., application of potassium fertilizer) is necessary to retain $K^+$. Comparison between two cultivars, drought-tolerant Zhongcha #108 and drought-sensitive Ruanzhi oolong, revealed that a stronger ability to remove reactive oxygen species (ROS), increase plasma membrane $H^+$- ATPase activity, and a negative membrane potential were the major factors that led to better $K^+$ retention in Zhongcha #108 [41]. In a drought-tolerant tea cultivar called Taicha #12, the effect of outwardly rectifying potassium levels on $K^+$ retention in mesophyll was relatively small, and non-selective cation channels (NSCC), which are activated by ROS, may have been the main path of $K^+$ leakage in mesophyll under drought conditions [42].

The research of Chen Linmu [43] also showed that stronger drought resistance correlates with higher $K^+$ content in tea leaves and better $K^+$ retention in mesophyll. Tran-

scriptome analysis revealed that under drought stress, the tea plant controls $K^+$ retention in mesophyll cells by regulating gene expression of potassium channels and potassium transporters. Considerable studies have proved that the high-affinity $K^+$ (HAK)/$K^+$ transporter (KT)/$K^+$ uptake (KUP) transporters, which are potassium transporters (CsHAKs), dominate $K^+$ acquisition and long-distance transport, especially under $K^+$ constraints. Analysis of specific problems and expression patterns induced by multiple stress types implied that CsHAKs participate in $K^+$ uptake and stress response in roots [44].

There is a mechanism behind $K^+$'s ability to alleviate drought stress in the tea plant: external supplementation of potassium, in association with chlorides and amino acids, relieves drought stress [45]. After further exploration, Xianchen [43] found that under drought conditions, the supply of $K^+$ causes lower $Cl^-$ outflow, which benefits osmotic balance in mesophyll cells.

### 3.4.2. Pest and Disease Resistance

Potassium deficiency results in increased susceptibility to infection, especially by fungi, which can be attributed to the role of potassium in primary metabolism and transport in the phloem of plants [46]. An adequate supply of potassium helps with pathogen tolerance, but in some cases, insects and necrotrophic pathogens are less likely to attack plants with potassium deficiency [47], which may be caused by low potassium induced increases in jasmonic acid and its derivatives, which act as a trigger in the system to defend against animals and insects [48].

Tea plants lacking potassium grow slowly, and are susceptible to Exobasidium vexans, anthrax, tea brown blight, etc. [49]. Increasing the application of potassium fertilizer can reduce the incidence of anthrax, Pestalotia theae and tea brown blight, and improve overall disease resistance. As for the reason behind improved resistance, it may be that a higher level of potassium facilitates the formation of organic substances such as phenolic compounds and peroxidases, which serve as phytotoxins in the tea plant to inhibit the growth, reproduction and spread of pathogens [50].

Sudoi [51] found that nitrogen–phosphorous–potassium (NPK) fertilizers can induce the tea plant's tolerance to mites. In the control of root-knot nematode in tea, Kamunya et al. [52] observed that the number of irregularly rounded roots decreased by 44% after two years of potassium treatment. Pruning combined with nitrogen and potassium application in a 1:2 ratio can greatly protect the tea plant from wormhole attack [53]. Spraying a mixture of urea and 1% sylvite can manage Cephaleuros parasiticus Karst, an algal pathogen that causes red rust disease [53]. Studies also showed that KCl or $K_2SO_4$ together with thiamethoxam or bifenthrin is more efficacious for the control of Helopeltis theivora than using insecticides alone [54].

### 4. Factors in Adjusting the Potassium Level in Soils

#### 4.1. Management of Potassium Fertilizers

During management of potassium fertilizers, the nutrient balance in soils must be considered to ensure high efficiency of potassium uptake. In China, the average available potassium in soils increased from 79.8 to 93.4 mg/L from 1990 to 2000. In more detail, the growth rates of cash crops were much faster than that of food crops, mainly because a higher dose of potassium fertilizer was applied to cash crops (1.4- to 2.6-fold more than to food crops) [55]. As the age of the plantation increased, five indicators increased: exchangeable potassium (EK), non-exchangeable potassium (NEK), potassium extracted from sodium tetraphenylboron, release threshold values of total labile potassium, and equilibrium potassium concentration [1].

Fertilizers containing potassium in all forms are a worthy addition to soils in tea gardens. $K_2SO_4$ [36,56,57] or KCl [21,22] can obviously raise the content of available potassium in soils in tea gardens, but the results of pot experiments are to the contrary. The reason may be that most of the available potassium is absorbed by tea in pot experiments [22]. Ruan et al. [7] considered that the optimal potassium dose of $K_2SO_4$ and KCl is between

124 and 160 kg/ha/y. Some scholars have shown that if the source of potassium is KCl, an NK ratio of 1:0.83 or 1:0.62 is best; if potassium is from $K_2SO_4$, an NK ratio of 1:0.21 or 1:0.42 is best [25,26].

Jianmei [36] ranked the effects of $K_2SO_4$ on the average content of available potassium in different soils: sandstone yellow soil > limestone yellow soil > acid purple soil. The results that Leqiong [22] obtained were slightly different: limestone yellow soil > sandstone yellow soil > acid purple soil. The performance of $K_2SO_4$ was better than KCl, but given budget limitations, KCl with a lower price can be put into use, too. It should be noted that extreme KCl supplementation will be harmful to the growth of the tea plant and even cause death of young trees' roots, so the dose of KCl must be carefully monitored [13].

### 4.2. Potassium-Solubilizing Bacteria (KSB)

Potassium deficiency in soils can be alleviated by potassium fertilizers, but the fact that potassium is an essential nutrient element for plants has promoted the production of potassium fertilizers worldwide, with commensurate rises in price. More expensive fertilizers accordingly lead to higher crop production costs that reduce farmers' profitability [58]. To overcome these limitations, a solution that transforms unavailable potassium into a usable form must be an alternative [59,60]. To our excitement, some soil microorganisms have been found to play a key role in the natural circulation and rapid dissolution of insoluble potassium-bearing minerals [61]. Uptake of available potassium after potassium-bearing minerals are dissolved with the assistance of those soil microorganisms may be more beneficial and economical than synthetic potassium fertilizers [62,63].

KSB bacteria extracted from the soil around the rhizomes of tea cultivars growing in Qimen County, Anhui Province, China were identified as Burkholderia sp. Inoculation of Burkholderia sp. can effectively increase the available potassium level in soils, which subsequently improves tea plant height and total polyphenol content [30]. KSB bacteria have a stronger solubilizing effect on unavailable potassium in growth media supplemented with KCl [64]. Tartaric acid and pyruvic acid produced by Bacillus are important components for potassium-solubilizing activity in vitro. Through pot experiments and in vitro enzymatic methods, it has been proven that organic acids secreted by inoculated bacteria can raise the level of available potassium in soils [29]. Bacillus pseudomycoides extracted from mica waste served as a biofertilizer to solubilize potassium, which led to an increase in available potassium in soils and enhanced plant yield [65,66].

### 5. Interaction between Potassium and Other Elements

Either single application of phosphorus fertilizers or excessive application of potassium fertilizers inhibits the absorption of nitrogen in soils by the tea plant, while proper single use of potassium fertilizers promotes this process. The carbon dioxide uptake of spring tea leaves can be increased in two ways: 1) applying NPK fertilizers only or 2) using both phosphorus and potassium fertilizers. In three samples treated with potassium fertilizers alone, the net photosynthetic rate decreased as the potassium application increased, but it was not radically different from the net photosynthetic rate of the unfertilized controls. Crush, tear, curl tea with different nitrogen concentrations had no difference in the contents of amino acids, which were relatively low. This result illustrates the importance of potassium in tea production [27].

KCl and $K_2SO_4$ can facilitate the growth of the tea plant, but chlorine in KCl has a negative effect: it reduces the activity of nitrate reductases in tea leaves, and decreases the content of free amino acids [39]. Ruan [7] and other researchers also showed that the highest dose of KCl (373 kg/ha) decreased tea yield slightly, which is a disadvantage of excessive KCl. By comparison, $K_2SO_4$ performed better in promoting the growth of the tea plant, as evidenced by sharp increases in biomass and amino acid content.

Potassium–magnesium fertilizers also benefit tea yield. Magnesium itself positively influences the levels of amino acids and caffeine in tea leaves, but using fertilizers with both magnesium and potassium raises their contents more dramatically [41]. Therefore,

the moderate use of both potassium and magnesium fertilizers is recognized as a key measure to improve tea productivity. It is appropriate to stabilize the ratio of potassium to magnesium ($K_2O$ to $MgO$) at 1:0.15~1:0.25 [12].

Potassium promotes nitrogen uptake in tea leaves of seedlings, inhibits phosphorus uptake in stems, and does not affect phosphorus uptake in roots and leaves. The research showed that [66] in hydroponic culture, there was a marked contrast in the absorption of minerals in various parts of seedlings under different concentrations of $K^+$: A high level of $K^+$ (20–40 mg/L) benefits the absorption of nitrogen, calcium, magnesium, and zinc in roots, but at the same time inhibits the accumulation of phosphorus, iron, copper, and manganese. An increase in $K^+$ concentration induces the accumulation of copper and zinc in stems, and nitrogen, magnesium, copper, and zinc in leaves, while inhibiting the accumulation of nitrogen, phosphorus, and magnesium in stems. However, a high level of $K^+$ (20–40 mg/L) is unfavorable for accumulating potassium, sodium, and phosphorus in leaves. After comparing effects of different potassium concentrations on the absorption of minerals by seedlings in sand culture, Changsong [67] observed that the copper and zinc contents in roots were positively correlated with the potassium concentration, while the absorption of manganese was negatively correlated. The results showed that the content of manganese in plants not treated with potassium was the highest in the leaves, and was much lower in stems and roots.

## 6. Conclusions

Along with nitrogen and phosphorus, potassium is the most important plant nutrient; it plays key roles in the growth and metabolism of the tea plant. Nowadays, over 59% of tea gardens in China are in a state of potassium deficiency, and the situation is exacerbated by a tea-picking period that lasts up to two-thirds of the year. There is some interaction between potassium and other elements including nitrogen, magnesium, copper, and zinc. Adequate potassium can improve the yield and quality of tea by activating and regulating diverse enzymes. In addition, it strengthens the resistance of tea plants to diseases, pests, and abiotic stresses. Moderate application of potassium fertilizers and KSB can increase the content of available potassium in tea gardens and maintain the balance of all forms of potassium in soils.

Traditionally, potassium fertilization strategies are designed to bring the soil to a target available potassium content, above which no increase in yield could be anticipated [9]. Therefore, Potassium fertilizer management is crucial. We suggest that further studies be conducted: (1) tracking the dynamic changes in potassium in soils after applying potassium fertilizers and/or KSB—quantifying the reserves of available K is important to designing appropriate fertilizer regimes and ensuring the sustainability of tea plantations grown; (2) Exploring combined effects of potassium and other elements—acquiring a deeper understanding of potassium in soils and the interactions between different elements will improve the management of potassium fertilizers in tea gardens, increase the utilization rate of potassium fertilizers, and maintain the balance of soil elements in the future; and (3) concentrating on use of some low-grade potassium minerals with suitable KSBs under field conditions, using different crops to develop an alternative and cheaper source of potassium in place of costly potassium fertilizers, which will be more appropriate in a specific farming system.

**Author Contributions:** Conceptualization, W.H.; methodology, W.H. and M.L.; validation, J.L. and M.L.; formal analysis, W.H. and A.L.; investigation, W.H.; resources, P.Z. and X.C.; data curation, W.H.; writing—original draft preparation, W.H.; writing—review and editing, W.T. and P.Z.; supervision, P.Z.; project administration, B.S., S.L. and P.Z.; funding acquisition, X.C. All authors have read and agreed to the published version of the manuscript.

**Funding:** This work was supported by the Science and Technology Project of Guangzhou (202102020290) and the Natural Science Foundation of Guangdong Province (2021A1515012091).

**Institutional Review Board Statement:** Not applicable.

**Informed Consent Statement:** Not applicable.

**Data Availability Statement:** Not applicable.

**Conflicts of Interest:** The authors declare no conflict of interest.

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
