# Peer review of "Effects of Potassium Deficiency on the Growth of Tea (Camelia sinensis) and Strategies for Optimizing Potassium Levels in Soil: A Critical Review"

_horticulturae, doi:10.3390/horticulturae8070660_

Round 1

Reviewer 1 Report

Title: Effects of potassium deficiency on the growth of tea (Camelia sinensis) and strategies for optimizing potassium levels in soil: a critical review.

 In this article, a well-written and well-structured review on potassium (K) nutrition in tea plants is presented, starting with the K dynamics in the soils, followed by the K uptake by the plant and the K metabolism in the tea plant, as well as the K effects on the quality of tea. The effect of abiotic and biotic stress on K dynamics is also mentioned and at the end, the authors address the issues on K management and the interaction with other relevant nutrients.

The paper sums up results from different articles. I believe that some additional work will be necessary to integrate the results presented by other authors. It would be very interesting to summarize the values in a final table that gather the information in the same units, which facilitate the reading and quick analysis of the main findings of optimal levels of K in soil for tea crop. Please also update the literature.

In conclusion, this is a very interesting paper due to the importance of tea crop and I recommend a revision of the manuscript. 

Reviewer 2 Report

The results were very interesting and corroborated with the literature. However, the authors left something to be desired in the discussion of the results, they could have explored the discussion further, emphasizing the importance of this study;

The work is interesting and compatible with the level of the paper;

All information about the work was understandable and consistent with each other;

Work the discussion further.

Reviewer 3 Report

A scientific publication "Effects of Potassium Deficiency on the Growth of Tea (Camelia 2 sinensis) and Strategies for Optimizing Potassium Levels in 3 Soil: A Critical Review" requires minor additions that I have highlighted in the manuscript and can be published in the scientific journal Horticulturae. 
